# Erosion Resistant Hydrophobic Coatings for Passive Ice Protection of Aircraft

**Naiheng Song *** and **Ali Benmeddour**

Aerospace Research Center, National Research Council of Canada, 1200 Montreal Road, Ottawa, ON K1A 0R6, Canada
* Correspondence: naiheng.song@nrc-cnrc.gc.ca; Tel.: +1-613-9988970

**Abstract:** Novel polymeric coatings, namely slippery polyurethane (SPU) coatings, with high surface hydrophobicity and superior erosion resistance against high speed solid particles and water droplets were successfully developed to protect the leading edge of fast moving aerodynamic structures, such as aircraft wings and rotor blades, against ice accretion. The coatings comprise newly synthesized surface-modifying polymers (SMPs) bearing fluorinated and polydimethylsiloxane branches at a loading level of 1–5 wt.%, based on the total resin solid, which showed good compatibility with the erosion-resistant polyurethane matrix (PU-R) and rendered effective surface hydrophobicity and slipperiness to the coatings, as evidenced by the high water contact angles of 100–115°. The coatings can be easily be sprayed or solution cast and cured at ambient temperature to provide highly durable thin coating films. X-ray photoelectron spectroscopy (XPS) investigation showed concentration of fluorine on the surface. The presence of 1–5 wt.% of SMPs in the polyurethane matrix slightly reduced the tensile modulus but had no significant impact on the tensile strength. All coating films exhibited good thermal stability with no material softening or degradation after heating at 121 °C for 24 h. DSC heating scans revealed no thermal transitions in the temperature range of −80 °C to 200 °C. Ice adhesion strength (IAS) tests using a static push rig in a cold room of −14 °C showed IAS as low as 220 kPa for the SPU coatings, which is much lower than that of PU-R (i.e., about 620 kPa). Sand erosion tests using 50 μm angular alumina particles at an impinging speed of 150 m/s and an impinging angle of 30° revealed very low erosion rates of ca. 100 μg/g sand for the coatings. Water droplet erosion tests at 175 m/s using 463 μm droplets with 42,000 impingements every minute showed no significant coating removal after 20 min of testing. The combination of the high surface hydrophobicity, low ice adhesion strength and superior erosion resistance makes the SPU coatings attractive for ice protection of aircraft structures, where the coatings' erosion durability is of paramount importance.

**Keywords:** icing protection; hydrophobic coating; surface modifying polymer; ice adhesion strength; erosion resistance



## 1. Introduction

Aircraft icing by super-cooled water droplets poses a major hazard for aviation [1]. For aircraft flying in cold regions, ice may form and accrete quickly on certain surfaces, particularly on the leading edge of a wing, the tail and the engine intakes, causing increased drag and weight and decreased thrust and lift, all of which are detrimental to aircraft performance. To mitigate the risks associated with icing, in addition to navigating away from the hazardous regions or grounding the aircraft [2,3], which is not always possible due to difficulties in accurately forecasting the hazardous freezing layers, commercial jets are commonly equipped with active anti-icing technologies, such as passing hot bleeding air from engine compressors through icing prone areas, forcing freezing point depressant fluid out of porous panels, and using electro-thermal ice protection systems or pneumatic de-ice boots [4]. However, despite the effectiveness of these systems, they are not applicable

to most small and light aircraft, such as helicopters, remotely piloted aircraft systems (RPAS) and next generation air taxis, due to system complexity, added weight and power requirement. As a result, intensive efforts have been made around the world to develop prospective icing protective measures, especially passive icephobic coatings that can minimize in-flight icing and ice adhesion and/or facilitate ice removal with minimal externally applied forces.

In the pursuit of effective icephobic coatings for aircraft ice protection, various coating designs have been explored, including superhydrophobicity, lubricant or freezing point depressant infusion, hydrated surfaces and low shear modulus coatings [5,6]. However, although numerous coatings have been developed to show substantially reduced ice adhesion strength (IAS), they all suffer critical shortcomings that limit their practical application. For example, superhydrophobic coatings are known to have low work of adhesion to liquid water; many studies [7,8] have shown that a superhydrophobic surface could delay water freezing, reduce IAS and facilitate thermal de-icing. However, their downsides are significant, including low resistance to condensation and frosting, poor mechanical integrity and difficulty in quality control and repair [9–11]. For lubricant- or freezing point depressant-infused surfaces, a very low IAS of <50 kPa has been achieved [12–14]. Nevertheless, such coatings have not shown the high mechanical robustness and fluids resistance required for aircraft applications. More importantly, the infused lubricant or freezing point depressant is susceptible to depletion due to the high-speed impact of water droplets as well as due to repeated icing/de-icing cycles [15]. The hydratable coatings take advantage of the strong coating–water interactions that keep a portion of water nonfreezing at sub-zero temperatures, thereby providing a water-lubricated slippery surface layer to depress ice formation and accretion [16–19]. However, the water up-take may swell and plasticize the coating, leading to reduced mechanical strength. Low shear modulus coatings (e.g., soft silicone elastomers) have recently drawn great attention due to their ability to afford extremely low IAS of less than 5 kPa [20–22]. Such low IAS values are highly attractive, but the high softness typically relates to high dust pick-up, deformation under air pressure and low erosion resistance against high-speed solid particles and rain droplets, making it difficult to use these types of materials for aircraft icing protection.

For a protective coating to be used on an aircraft, in addition to high icephobicity, it must meet comprehensive property requirements, such as workability, heat and low temperature resistance, weatherability, erosion durability and reparability. Among these, erosion resistance is the most important, since the coating applied to the leading edge of a wing or helicopter rotor blades, where icing tends to occur and cause the most negative impact on aircraft operation, may experience excessive wear due to the high-speed (e.g., 100–300 m/s) impact of sand, airborne dust, rain droplets and hail. To mitigate erosion damage, shield materials such as metal strips or caps made of nickel, titanium and stainless steel, as well as polymeric erosion protective tapes, are typically applied to the leading edges. Although studies have been carried out to understand the environment and surface properties in relation to IAS, and ceramic erosion-resistant coatings have been investigated for icing protection [23,24], the effectiveness of metals and ceramic coatings is limited due to their low ice adhesion reduction factor (e.g., <2) and poor applicability on large curved surfaces. In addition, erosion resistant polymeric coatings with reduced ice adhesion strength remain unknown.

Thus, it is the objective of this study to develop highly robust icephobic polymer coatings suitable for aircraft ice protection at leading edges, which have not only high icephobicity and superior erosion resistance, but also excellent thermal and mechanical properties, along with easy applicability on large curved surfaces. To this end, surface modification of a highly erosion resistant coating, namely PU-R, was carried out by introducing new surface-modifying polymers (SMPs) into the coating formulation, thereby producing novel slippery polyurethane (SPU) coatings with high surface hydrophobicity. The PU-R is a high solid, 2-part, solvent-borne aliphatic polyurethane coating developed at the National Research Council Canada. The coating demonstrated superior erosion resistance against

both high-speed solid particles and rain droplets, outperforming commercial aircraft top-coat and erosion protective tapes [25]. It is expected that the resulting SPU coatings could retain the exceptional erosion resistance of PU-R while exhibiting improved icephobicity to mitigate the hazards caused by icing on the leading edge surfaces of aircraft wings and helicopter rotor blades.

## 2. Materials and Methods

### 2.1. Materials

Desmodur® Z4470 BA (Z4470, NCO% = 11.9%, functionality ~3.5, Covestro, Leverkusen, Germany), anhydrous N,N-dimethylformamide (DMF, Sigma-Aldrich, St. Louis, MO, USA), 1H,1H,2H,2H-perfluoro-1-octanol (PFOA, Career Henan Chemicals, Zhengzhou, China), Terathane® PTMEG-650 (average Mw of 650 g/mol, Invista, Wichita, KS, USA), MCR-C18 (monocarbinol terminated polydimethylsiloxane, average Mw of 5000 g/mol, Gelest Inc., Bucks County, PA, USA) and dibutyltin dilaurate (DBTDL, Sigma-Aldrich) were used in the as-received conditions. Fiberglass plates (FR-4, $2 \times 2 \times 1/4$ in) were purchased from Curbell plastics. Aeordur HS 2118 CF primer (Akzo Nobel, Amsterdam, The Netherland) was purchased from Hypercoat-Downing Ltd., Mississauga, ON, Canada, X-9032/G401 Nix Stix® release agent was acquired from Stoner Molding Solutions. The matrix coating, namely PU-R, is a 2-part solvent-borne polyurethane elastomer coating developed at the National Research Council Canada for erosion protection of helicopter rotor blades against high-speed solid particles and rain droplets. Its formulation will be published elsewhere.

### 2.2. Methods

For property studies, coating and free-standing thin film samples were prepared by casting the coating solutions onto a fiberglass substrate (FR-4, $2 \times 2 \times 1/4$ in) and into an aluminum mold ($4.5 \times 4.5$ in) pre-treated with X-9032/G401 Nix Stix® release agent, respectively, followed by drying and curing at ambient conditions. Through controlling the amount of coating solutions cast, all the coatings on FR-4 and free-standing thin films have comparable thicknesses of ca. 0.35 mm.

Differential scanning calorimetry (DSC) was performed on DSC Q2000 (TA Instruments, New Castle, DE, USA) in nitrogen at specific heating and cooling profiles. FT-IR spectroscopy was recorded on Nicolet 6700 (Thermo Fisher Scientific, Waltham, MA, USA) equipped with Smart iTR™ attenuated total reflectance (ATR) sampling accessory. Test samples were measured in the form of thin films that were pressed against a diamond plate with single bounce mode. The complex viscosity ($\eta$) of the coatings were measured by a DHR-2 rheometer (TA Instrument Co.) at 25 °C using a cone-plate geometry (40 mm, 0.9786°). The strain was set to be 5.85% and the angular frequency was set to be 10 rad/s. The X-ray photoelectron spectroscopy (XPS) data were collected using a Kratos AXIS Ultra DLD spectrometer with a monochromated Al K-alpha beam (1486.6 eV) under high vacuum ($5 \times 10^{-9}$ Torr). The binding energy scale was charge corrected by shifting the main peak of the C 1s spectrum to 284.8 eV. The electron collection lens aperture was set to sample a 700 μm × 300 μm spot, the largest possible, and at least two spots were measured for each sample to ensure the compositional information was representative of the average for the surface. The charge neutralizer was used in all measurements because the samples are non-conductive. The data were analyzed using CasaXPS (version 2.3.17PR1.1).

Tensile properties of thin films were measured on Instron model 5565 tensile tester equipped with pneumatic grips according to standard ASTM D412. Dumbbell-shaped film coupons were die cut using a DIN-53504-S3A type cutting die (ODC tooling & molds). All samples were conditioned at $23 \pm 2$ °C and $50 \pm 5\%$RH for at least 24 h before testing. Due to the fact that slippage at the grip areas occurred during testing, benchmarks of $10 \pm 1$ mm distance (LO) in the middle of the dumbbell-shaped samples were drawn and followed during testing to obtain true elongation at break. The rate of grip separation was 500 mm/min. After rupture, the distance (LF) between the benchmarks was measured

within ca. 1 min for the calculation of tensile set. The Shore A hardness of the thin films was measured using a Rex DD-4 digital durometer with an OS-1 operating stand.

Solid particle erosion tests were performed according to ASTM Standard G76-04 using angular alumina particles of ca. 50 μm (AccuBRADE 50, part No.: AP106, S.S. White Technologies, Petersburg, FL, USA) as the erodent. During testing, the alumina particles were fed into a compressed air carrier stream from a pressurized vibrator-controlled hopper, which was allowed to pass through a silicon carbide nozzle with an inner diameter of 1.14 mm and directed towards the test sample at a preset impingement angle of 30° with respect to the test sample surface. The impingement speed of the ejected alumina particles was 150 m/s, which was controlled through adjusting the pressure of compressed air. The particle flux was regulated at 3–5 g/min by changing the vibrating amplitude of the hopper. The stand-off distance was maintained constant at 50 ± 1 mm. After each 4 min of testing, the test sample was removed from the erosion rig and measured for its weight using an analytical balance with an accuracy of ± 0.01 mg. At the same time, the weight of the consumed erodent was measured. Then the sample was returned to the test rig and erosion testing was resumed. The erosion rates were calculated by dividing the mass loss of the coatings with the mass of solid particles consumed. The maximum particle loads were estimated by dividing the amount of sand required to fail the coating sample (e.g., penetration) with the eroded area (i.e., approximately a circle of ca. 10 mm diameter).

The water droplet erosion (WDE) resistance of the coated samples were evaluated according to ASTM Standard G73 using a testing rig that was equipped with a working chamber, a vacuum system, a compressed air driven turbine and a water droplet generating system. During testing, two test coated coupons having mass differences of less than 0.1 g were mounted on the opposite ends of a rotating disc, with one as the comparative control and the other as the test sample. The disc was rotated at a specific speed while water droplets were formed in the test chamber on a path of the test coupons. A 30–50 mbar vacuum was maintained during the test to avoid temperature rise caused by friction between the rotating disc and air. In this study, the disc was rotated at a rate of 7000 rpm, corresponding to a water droplet impingement velocity of 175 m/s. The impingement angle was 90°. The average size of water droplets produced using a 400 μm shower head was about 463 μm. The test coupon underwent about 42,000 individual water droplet impingements during each minute of testing. For the test coupon preparation, AA2024 specimens (0.32″ × 0.97″ × 0.12″) were first treated using Akzo Nobel's Metaflex® SP 1050 pretreatment, followed by priming using Aerodur HS 2118 primer (ca. 30 μm thick). The coatings of this study were applied by dispensing ca. 0.10 g of the coating solutions on the primed surface, followed by drying and curing at ambient conditions for 7 days. The coating layers had an average thickness of ca. 0.35 mm.

Static and dynamic water contact angles (CAs) were measured on an Automated Goniometer (Model 290, Ramé-hart Instrument Co., Succasunna, NJ, USA). For static contact angles, a water droplet of 4 μL was dispensed on the coating surface using an automated dispenser. The contact angles were analyzed by the Dropimage Advanced software. The measurement of dynamic contact angles, i.e., the advancing and receding angles, was performed using the volume addition and subtraction method, with each volume step being 2 μL and a delay time of 0.5 s. The maximum contact angle during volume addition was taken as the advancing angle, while the contact angle where the drop edges started to slide during volume subtraction was recorded as the receding angle.

Ice adhesion of the coatings was examined using the push-rig schematically shown in Figure 1 [26]. The set-up comprises a "pusher" shaft connected to a load cell, which, in turn, is connected to a threaded rod that provides lateral movement of the assembly into the test sample. The threaded rod was manually turned to drive the pusher against the ice block enclosed by a PTFE lined aluminum mold. Output from the load cell was recorded and the maximum force required to detach the ice block from the coating surface is used to calculate IAS. The aluminum mold has a dimension of 3″ × 3″ × 5/16″, the PTFE has a thickness of 3/8″ and the cavity for ice block preparation is 1″ × 1″ × 5/16″. To prepare

the test sample, the PTFE lined aluminum mold was clamped tightly to the coating surface, with a rubber cushion (1/16″ thick) sandwiched between the two. De-ionized water was filled into the cavity and the assembly was then kept in a cold room at −14 °C to freeze the water overnight. Figure 1 also shows the PTFE-lined aluminum mold and the preparation of the ice cube on the coating surface. Both the test sample preparation and the push rig test were performed in a cold room that was kept constantly at −14 °C.

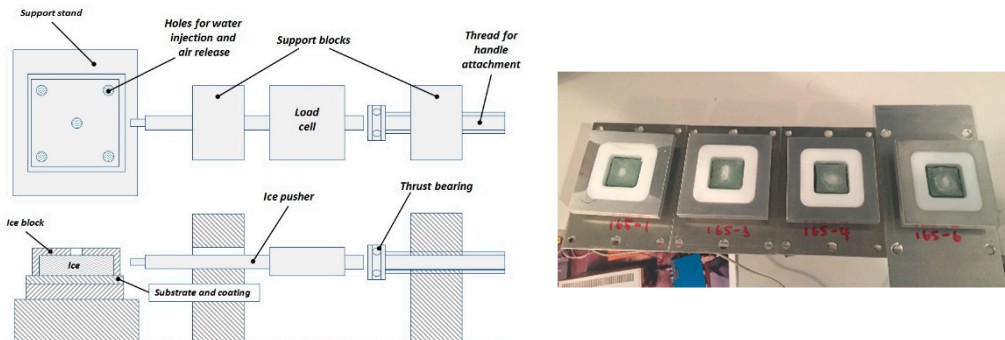

**Figure 1.** Schematic drawings of the push rig for ice adhesion test (**left**) and a picture of test specimens with ice formed in a cavity of PTFE-lined aluminum mold on top of coating surfaces (**Right**).

*2.3. Preparation of Surface-Modifying Polymers (SMPs) and Durable Hydrophobic Polyurethane (SPU) Coatings*

Synthesis of SMP-1: 1H,1H,2H,2H-perfluoro-1-octanol (PFOA, 6.5 g) and dibutyltin dilaurate (DBTDL, 0.3 g) were added to a solution of Desmodur® Z4470 BA (10.0 g) in 25 mL of anhydrous N,N-dimethylformamide (DMF). The mixture was heated to 60 °C and stirred under nitrogen for 2 h before Terathane® PTMEG-650 (3.7 g) in 10 mL of anhydrous DMF was added, dropwise. The resulting reaction mixture was stirred at 60 °C for another 3 h. After cooling to room temperature, the viscous reaction solution was precipitated in 300 mL of de-ionized water/methanol (4/1, *v/v*). The sticky solid formed was washed twice with warm de-ionized water (300 mL) and once with methanol (200 mL), then dried at 75 °C in a convection oven for 72 h to yield 16.0 g of a clear rubbery product (yield 80%).

Synthesis of SMP-2: PFOA (6.2 g), MCR-C18 (4.1 g) and DBTDL (0.5 g) were added to a solution of Desmodur® Z4470BA (10.0 g) in 30 mL of anhydrous DMF. The mixture was heated to 60 °C and stirred under nitrogen for 4 h before Terathane® PTMEG-650 (3.7 g) in 10 mL of anhydrous DMF was added. The reaction mixture was stirred at 60 °C for another 3 h. After cooling to room temperature, the viscous reaction solution was precipitated in 300 mL of de-ionized water/methanol (4/1, *v/v*) and the resulting sticky solid was washed twice with warm de-ionized water (300 mL) and once with methanol (200 mL), then dried at 80 °C in a convection oven for 72 h to yield 21.6 g of an opaque gel product (yield 90%).

Synthesis of SMP-3: PFOA (5.9 g), MCR-C18 (8.1 g) and DBTDL (0.5 g) were added to a solution of Desmodur® Z4470BA (10.0 g) in 30 mL of anhydrous DMF. The mixture was heated to 60 °C and stirred under nitrogen for 4 h before Terathane® PTMEG-650 (3.7 g) in 10 mL of anhydrous DMF was added. The reaction mixture was stirred at 60 °C for another 3 h. After cooling to room temperature, the viscous reaction solution was precipitated in 300 mL of de-ionized water/methanol (4/1, *v/v*) and the resulting sticky solid was washed twice with warm de-ionized water (300 mL) and once with methanol (200 mL), then dried at 80 °C in a convection oven for 72 h to yield 23.8 g of an opaque gel product (yield 86%).

Preparation of SPU coating and thin film samples: SPU coating solutions were prepared by formulating the SMPs (i.e., SMP-1 to SMP-3) into the erosion-resistant polyurethane coating PU-R at levels of 1.0, 3.0 and 5.0 wt.%, respectively, based on the total resin solid. The coating solutions (ca. 1.50 g) were cast on fiberglass substrates (FR-4, 2 × 2 × 1/4 in), followed by drying and curing at ambient conditions for 7 days to produce coatings with a dry film thickness of ca. 0.35 mm. The coating samples for water droplet erosion tests were prepared by casting ca. 0.10 g of coating solution on aluminum specimen (Al 2024 alloy,

0.31 × 0.85 × 0.12 in), pre-coated with a chromate-free Aeordur HS 2118 CF primer (ca. 40 μm thick). The coatings were dried and cured at ambient conditions for 7 days before testing. For thin film samples, ca. 7.62 g of coating solutions were cast into an aluminum mold (4.5 × 4.5 in) pre-treated with X-9032/G401 Nix Stix® release agent. The coating solutions were allowed to flow and level naturally. After drying and curing at room temperature for 1 day, the resulting thin films (ca. 0.35 mm thick) were removed from the mold and allowed to further cure at ambient conditions for 6 more days before testing.

## 3. Results and Discussion

### 3.1. Synthesis of SMPs and SPU Coatings

To mitigate icing and ice accretion on the leading edge surfaces of aircraft wings and helicopter rotor blades without sacrificing erosion durability against high-speed solid particles and rain droplets, it would be straightforward and ideal to modify the existing erosion resistant coatings to render the surface with icephobicity. To this end, effort was first made by introducing various commercial surface modifying agents such as polyether-modified polydimethylsiloxane (e.g., BYK-306, BYK), fluorinated copolymers (e.g., CapstoneTM FS-83, Chemours), hydrophobic fumed silica (e.g., Aerosil® R 812, Evonik), PTFE micropowders (e.g., Zonyl® MP 1100, Chemours) and PEGylated polydimethylsiloxane (e.g., DBE-224, Gelest) into the erosion resistant PU-R coating at different levels in hoping to achieve hydrophobic and slippery surfaces that can delay ice formation or reduce ice adhesion strength. Unfortunately, limited success was achieved. None of these additives led to coatings with high surface hydrophobicity. Furthermore, foam stabilization and phase separation were observed when some of the additives (e.g., FS-83 and DBE-224) were incorporated at high amounts (e.g., >1 wt.%, based on total resin solid). Thus, to achieve durable icephobic coatings for leading edge icing protection, new surface-modifying polymers (SMP-1 to SMP-3) were designed and synthesized in this study, which bear low surface tension fluorinated and polydimethylsiloxane (PDMS) branches, and have good compatibility with the polyurethane matrix, to enable high PU-R surface hydrophobicity.

Scheme 1 shows the synthetic scheme of the SMPs, where the multifunctional polyisocyanate Z4470 BA (f ≈ 3.5) first reacts with the mono hydroxyl compounds (i.e., PFOA and MCR-C18) to produce a reaction mixture, followed by polymerizing with the diol PTMEG-650. The intermediate reaction mixture consists of a distribution of reaction products of Z4470 with the mono hydroxyl compound (e.g., unreacted, mono-, di-, tri- and fully reacted Z4470), depending on the molar ratio of Z4470/(PFOA+MCR-C18). Structurewise, it is preferred to have a minimum amount of unreacted and fully reacted Z4470 in the intermediate reaction mixture. The former has high functionality and tends to lead to crosslinking when polymerizing with PTMEG-650, while the latter may contribute to incompatibility of SMP with matrix PU-R due to poor inter-chain interactions. At the same time, the di-reacted Z4470 product that has a remaining isocyanate functionality of about 2 should be maximized so as to produce high molecular weight SMPs with branched structures. For the reaction system of this study, it was experimentally found that a suitable molar ratio of Z4470/(PFOA+MCR-R18) was about 1/2.2, where highly branched polymers could be obtained from the subsequent polymerization reaction with PTMEG-650 without crosslinking.

**Scheme 1.** Synthetic scheme of the surface-modifying polymers.

Table 1 summarizes the syntheses of the SMPs using different combinations of PFOA and MCR-C18 as the mono hydroxyl compounds for the purpose of examining the effects of fluorinated branches and silicone branches of the SMPs on coating properties. The reaction of N4470 with PFOA and MCR-C18 at 60 °C in the presence of DBTDL as a catalyst gave clear solutions, although MCR-C18 was initially immiscible with the reaction mixture. The addition of PTMEG-650 quickly increased the solution viscosity, suggesting the build-up of molecular weight. FTIR investigations showed the disappearance of the characteristic NCO absorption at around 2265 cm$^{-1}$ in about 2 h of reaction time, indicating the completeness of the reactions. SMP-1 was obtained as a clear rubbery gel, while SMP-2 and SMP-3 were white soft gels. The white color of SMP-2 and MSP-3 may be attributed to the phase separation of PDMS segments within the SMPs. Theoretical calculations based on the stoichiometry ratios gave a residual hydroxyl content in SMPs in the range of 0.029–0.040 mmol/g, which is small and not expected to affect the stoichiometry of PU-R coating formulation.

**Table 1.** Syntheses of SMPs.

| SMPs | Molar Equivalence | | | | PFOA (wt.%) | MCR-C18 (wt.%) | Residual OH (mmol/g) |
|------|------|------|--------|-----------|-------------|----------------|----------------------|
|      | Z4470 | PFOA | MCR-C18 | PTMEG-650 | | | |
| SMP-1 | 1 | 2.2 | - | 0.7 | 31.8% | 0.0% | 0.040 |
| SMP-2 | 1 | 2.1 | 0.1 | 0.7 | 25.7% | 16.8% | 0.034 |
| SMP-3 | 1 | 2.0 | 0.2 | 0.7 | 21.1% | 29.0% | 0.029 |

*3.2. Coating Formulation and Preparation*

To prepare coatings with high surface hydrophobicity and excellent erosion durability, the SMPs were formulated into the 2-part solvent-borne PU-R coatings at loading levels of ca. 1 wt.%, 3 wt.% and 5 wt.%, respectively, based on the total resin solid (Table 2). All the coating solutions (i.e., PU-R, SPU-1 to SPU-3) are clear, homogenous and free of bubbles. They have comparable solid contents of ca. 58 wt.% and complex viscosities of ca. 200–230 cp. Pot life was about 4 h, when the viscosity increased by 50%. The coatings can be easily applied using a low volume medium pressure (LVMP) spray gun or by solution casting. The tack-free times, when the coating surfaces became non-sticking to a touching spatula, were about 4 h. Although the FTIR spectra of the thin film samples indicated that curing was complete after 2 days (Figure 2A), as evidenced by the disappearance of characteristic NCO absorption peak at around 2240 cm$^{-1}$, all coating and thin film samples were allowed to cure at ambient conditions (e.g., 21 ± 2 °C, 20–80% relative humidity) for 7 days before property testing. Tough and elastomeric SPU dry films with a Shore A hardness of about 70 were obtained after the curing. Both PU-R and SPU coatings showed good adhesion to FR-4. No coating removal could be made without damaging the fiberglass substrate.

For property studies, coating and free-standing thin film samples were prepared by casting the coating solutions onto a fiberglass substrate (FR-4, 2 × 2 × 1/4 in) and into an aluminum mold (4.5 × 4.5 in) pre-treated with X-9032/G401 Nix Stix$^{®}$ release agent, respectively, followed by drying and curing at ambient conditions.

**Table 2.** Preparation and water contact angle measurements of SPU coatings.

| Coatings | SMPs | wt.% of SMPs | θ, Water | θ$_{adv}$, Water | θ$_{rec}$, Water |
|---|---|---|---|---|---|
| PU-R | - | 0 | 69.0 | 77.4 | 20.4 |
| SPU-1_1% | | 1.0% | 103.7 | 108.0 | 80.2 |
| SPU-1_3% | SMP-1 | 3.0% | 109.7 | 115.5 | 82.1 |
| SPU-1_5% | | 5.0% | 109.6 | 114.0 | 82.0 |
| SPU-2_1% | | 1.0% | 100.0 | 101.2 | 78.3 |
| SPU-2_3% | SMP-2 | 3.0% | 109.0 | 114.0 | 80.5 |
| SPU-2_5% | | 5.0% | 109.3 | 115.9 | 79.4 |
| SPU-3_1% | | 1.0% | 96.9 | 100.7 | 69.4 |
| SPU-3_3% | SMP-3 | 3.0% | 109.5 | 114.8 | 78.8 |
| SPU-3_5% | | 5.0% | 112.0 | 117.5 | 81.5 |

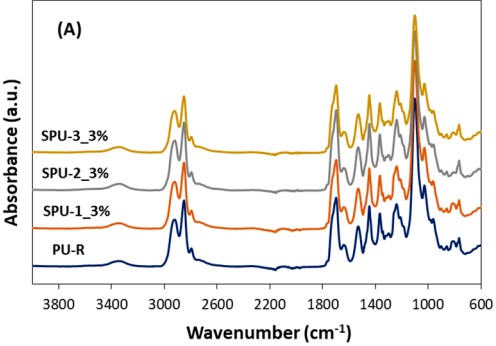 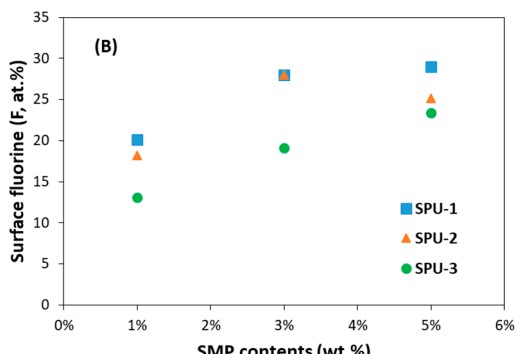

**Figure 2.** (**A**) FTIR spectra of PU-R and SPU coating films after curing at ambient conditions for 2 days, and (**B**) surface fluorine content (at.%) of SPU coatings from XPS studies.

### 3.3. Properties of SPU Coatings

Table 2, above, displays the static and dynamic water contact angles of the coatings, measured using a goniometer equipped with an automated dispenser. As expected, the unmodified matrix PU-R is hydrophilic and has a static water contact angle (θ) of 69°, whereas all SPU coatings comprising the SMPs are hydrophobic, with θ being larger than 97°. Water beaded up quickly and slid easily on the coating surfaces. Different SMPs showed no substantial differences in producing hydrophobic surfaces. However, SPU-2 and SPU-3 did give a more slippery feel than SPU-1 when rubbing fingers on the coating, probably due to the presence of PDMS on the surface. No smudge was observed after the finger rubbing, and the surface hydrophobicity could be easily restored for oil-contaminated surfaces by washing with detergent. A dependence of the water contact angle on the SMP content was revealed, as evidenced by the increase in θ when the SMP loading was increased from 1 wt.% to 3 wt.%, based on the total resin solid. Further increasing the SMP loadings to 5 wt.% did not lead to additional increase of θ. The hydrophilicity of PU-R was more clearly seen from its small receding angle (θ$_{rec}$) of ca. 20° and large contact angle hysteresis (CAH) of ca. 57°. In contrast, all SPU coatings exhibited high θ$_{rec}$ of ca. 80° and small CAH in the range of 22–36°. Based on the relationship between water wettability and ice adhesion proposed by Muller et al. [27], as illustrated by Equation (1), where τ$_{ice}$ is the ice adhesion strength, the matrix PU-R is expected to have an IAS of ca. 650 kPa, while the SPU coatings should have reduced IAS of ca. 400 kPa.

$$\tau_{ice} = (340 \pm 40 \text{ kPa})(1 + \cos\theta_{rec}), \tag{1}$$

To understand the surface elemental compositions, XPS was recorded for PU-R and the SPU coatings. As shown in Table 3, all the SMP-containing SPU coatings showed a high

fluorine content in the range of 13–29 at.%, indicating the enrichment of SMPs on coating surfaces. In contrast, the PU-R showed no fluorine on the surface. Figure 2B compares the surface fluorine contents of the SPU coatings. It can be seen that SPU-1 coatings have the highest fluorine contents, compared to SPU-2 and SPU-3, due to the high PFOA content in SMP-1. SPU-3 coatings, which are based on SMP-3, showed the lowest surface fluorine contents. For SPU-1 and SPU-2, the fluorine content seems to reach a plateau at a SMP loading of 3 wt.%, which agrees with the water contact angle measurement, where an increase of static water contact angle occurred from SPU-1_1% and SPU-2_1% to SPU-1_3% and SPU-2_3%, respectively, and leveled off thereafter. The SPU-3 coatings showed a monotonous increase of fluorine content with the SMP-3 loadings. For SPU-2 and SPU-3 coatings, it was expected that the silicone branches would behave similarly to the fluorinated chains in terms of surface enrichment. However, due to unknown reasons, XPS analysis of the Si content did not show a clear trend of increasing Si with the increasing MCR-C18 content of SMP. The detection of Si for PU-R and SPU-1 coatings, where no MCR-C18 was used, may be attributed to the presence of a silicone-containing surface additive, namely BYK-306 (BYK Additives and Instruments), in the PU-R formulation at a loading level of ca. 0.3 wt.%, based on total coating formulation. The decreasing Si content from PU-R to SPU-1 and to SPU-2 may be attributed to the strong tendency of SMP-1 to migrate to the surface during the drying and curing processes, which suppressed the enrichment of BYK-306 at the surface.

**Table 3.** XPS analysis of PU-R and SPU coatings.

| Coatings | Elemental Content (at.%) | | | | |
|---|---|---|---|---|---|
| | C 1s | N 1s | O 1s | F 1s | Si 2p |
| PU-R | 69.38 | 2.79 | 20.79 | 0 | 7.04 |
| SPU-1_1% | 59.11 | 4.31 | 13.14 | 20.08 | 3.37 |
| SPU-1_3% | 56.66 | 4.96 | 9.56 | 27.96 | 0.87 |
| SPU-1_5% | 56.16 | 4.87 | 9.18 | 28.95 | 0.85 |
| SPU-2_1% | 58.56 | 4.00 | 14.28 | 18.21 | 4.95 |
| SPU-2_3% | 56.83 | 4.73 | 9.41 | 27.95 | 1.08 |
| SPU-2_5% | 56.31 | 4.49 | 11.02 | 25.16 | 3.03 |
| SPU-3_1% | 58.76 | 3.39 | 16.91 | 13.05 | 7.89 |
| SPU-3_3% | 56.05 | 3.64 | 14.05 | 19.07 | 7.19 |
| SPU-3_5% | 56.00 | 4.24 | 11.92 | 23.36 | 4.48 |

All SPU coatings exhibited good thermal stability with no material softening or degradation after heating at 121 °C for 24 h. Figure 3 shows the representative second DSC heating scans of the coatings recorded in nitrogen at a heating rate of 20 °C/min, which revealed no thermal transitions in the temperature range of −80 °C to 200 °C. These coating samples were pre-treated by heating to 200 °C at 20 °C/min, cooling to −50 °C at a rate of 10 °C/min, and stabilized at −80 °C before the second heating. Neither the first heating nor the cooling scans showed discernible glass or melting transitions, indicating thermoset properties for the coatings and high heat resistance.

Table 4 summarizes the tensile properties of the thin film samples, measured according to ASTM Standard D412 at a grip separation rate of 500 mm/min. It can be seen that all coating films exhibited high tensile strength (e.g., 19–27 MPa), high elongation at break (e.g., 640–730%) and low tensile set (e.g., 20–35%). In comparison with the matrix PU-R, the incorporation of SMP-1 did not substantially affect the tensile strength (SPU-1 in Figure 4A), but a slight decrease in modulus, as indicated by stress at 500% strain, was observed (Figure 4B). For SPU-2 and SPU-3, marginally decreased tensile strength and modulus were noticed with the increasing amount of SMP-2 and SMP-3, respectively. Elongation at break increased from 640% for PU-R to 680–730% for SPUs, depending on the SMP content, but no sizable changes in tensile set were observed. The small tensile sets (i.e., 20–35%) recorded within 1 min of sample failure indicate high resilience of the coating films.

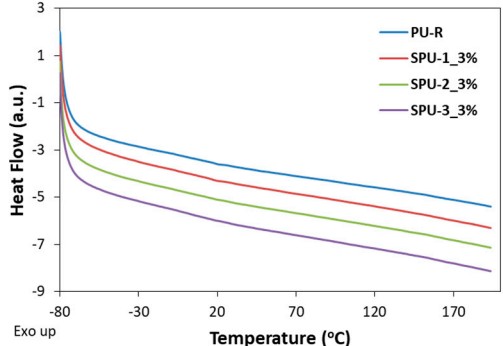

**Figure 3.** Representative second DSC heating scans of PU-R and SPU coatings in nitrogen at a heating rate of 20 °C/min.

**Table 4.** Tensile properties of PU-R and SPU coating films.

| | Tensile Strength (MPa) | Stress at 300% Strain (MPa) | Stress at 500% Strain (MPa) | Elongation at Break (%) | Tensile Set (%) |
|---|---|---|---|---|---|
| PU-R | 26.8 | 5.4 | 12.2 | 640% | 25% |
| SPU-1_1% | 27.2 | 5.2 | 11.2 | 680% | 20% |
| SPU-1_3% | 26.7 | 5.2 | 11.0 | 700% | 20% |
| SPU-1_5% | 23.9 | 4.9 | 9.9 | 720% | 30% |
| SPU-2_1% | 25.2 | 5.2 | 11.3 | 690% | 20% |
| SPU-2_3% | 18.9 | 4.5 | 8.9 | 690% | 20% |
| SPU-2_5% | 20.1 | 4.2 | 8.5 | 730% | 30% |
| SPU-3_1% | 23.0 | 5.1 | 10.7 | 690% | 25% |
| SPU-3_3% | 22.9 | 5.0 | 10.6 | 700% | 35% |
| SPU-3_5% | 18.7 | 4.5 | 8.8 | 700% | 30% |

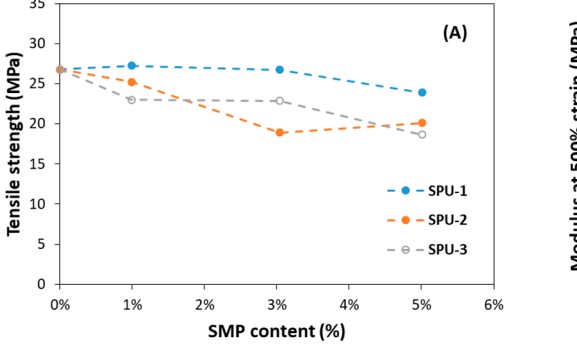
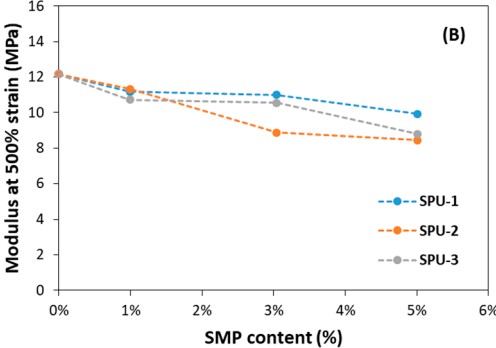

**Figure 4.** Plots of tensile strength (**A**) and modulus (**B**), as indicated by stress at 500% strain of SPU coatings against the SMP contents.

Figure 5A presents the IAS of the SPU coatings measured at −14 °C. All the detachment occurred at the ice-coating interface due to adhesion failure, no breakage of ice block was observed. The matrix PU-R showed an IAS of about 622 kPa and the SPU coatings showed IAS in the range of 220–480 kPa. No substantial differences in IAS were discovered between the SPU coatings, regardless of the different SMPs and varying SMP loadings, except for SPU-3, which exhibited some lower IAS (e.g., ca. 250 kPa) at SMP-3 loadings of ≥3 wt.%. However, as expected, all these IAS values are in the range of hydrophobic fluorinated polymers (e.g., 150–350 kPa for PTFE [28]) and silicones (200–400 kPa [20]) and agree well with the predicted values according to Equation (1) by Muller et al. [27]. Figure 5B displays the IAS test results for coating samples that were subjected to repeated icing/ice removal cycles. As it can be seen, the IAS of matrix PU-R fluctuated in the range of 606–715 kPa

while the IAS of SPU-3_3% varied in the range of 220–370 kPa, which are in the error range based on Equation (1). The fact that no substantial increase in IAS was observed for SPU-3_3% indicates a good stability against multiple icing/ice removal cycles.

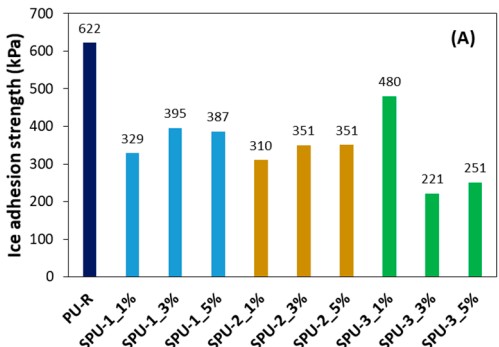 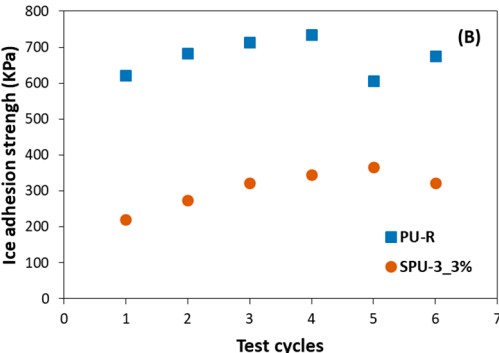

**Figure 5.** Ice adhesion strength (in kPa) of SPU coatings measured using a static push rig at −14 °C (**A**) and ice adhesion strength of PU-R and SPU-3_3% coatings that were subjected to repeated icing/ice removal cycles (**B**).

Although hydrophobic PTFE and commercial silicones showed reduced ice adhesion strength, they were found to have neither good erosion durability against high-speed particles nor good applicability for aircraft leading edge protection applications. To address these shortcomings, the erosion durability of the SPU coatings against both solid particles and water droplets were evaluated according to ASTM Standards G76 and G73, respectively.

Figure 6 shows the solid particle erosion rates and maximum particle loads of the matrix PU-R and representative SPU coatings (i.e., SPU-1_3% and SPU-2_3%), by blasting the coatings with angular alumina particles of 50 μm at an impinging velocity of 150 m/s and an impinging angle of 30°. Figure 6 compares the erosion rates and maximum particle loads between the coatings. It can be seen that both SPU-1_3% and SPU-2_3% exhibited low erosion rates of about 130–140 μg/g sand and high particle loads of 66–73 g/cm². In comparison with the matrix PU-R, only a slight increase in erosion rate was observed for the SPU coatings due to the incorporation of SMPs; the maximum particle loads remained substantially unchanged. As comparative references, commercial aircraft exterior coatings such as Alumigrip 4200 polyurethane topcoat (Akzo Nobel) and Aeroglaze® 1433 polyurethane midcoat (Socomore) were subjected to the same erosion test and failed, with extensive coating removal and substrate exposure at particle loads of <20 g/cm².

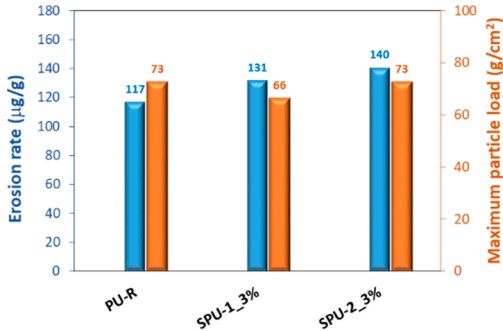

**Figure 6.** Sand erosion resistance of PU-R and SPU coatings. Test conditions: 50 μm angular alumina particles, impact velocity of 150 m/s, impact angle of 30°.

The water droplet erosion (WDE) resistance of the SPU coatings was evaluated using a water spin rig test facility, at an impact velocity of 175 m/s and an impingement angle of 90°. The coating configuration is Metaflex® SP 1050 pretreatment/Aerodur HS 2118 primer (ca. 30 μm)/PU-R or SPU (ca. 0.35 mm). Figure 7 presents the pictures of the coatings

after certain periods of WDE testing. In comparison with commercial Alumigrip 4200 polyurethane topcoat, where total material removal was observed at the impact area (a line of ca. 2 mm wide) at 2.5 min of testing, the PU-R an SPU coatings showed only minimal material loss at the edges even after 20 min of testing, indicating excellent water droplet erosion resistance of the coatings. The introduction of SMPs did not substantially affect the water droplet erosion resistance of PU-R.

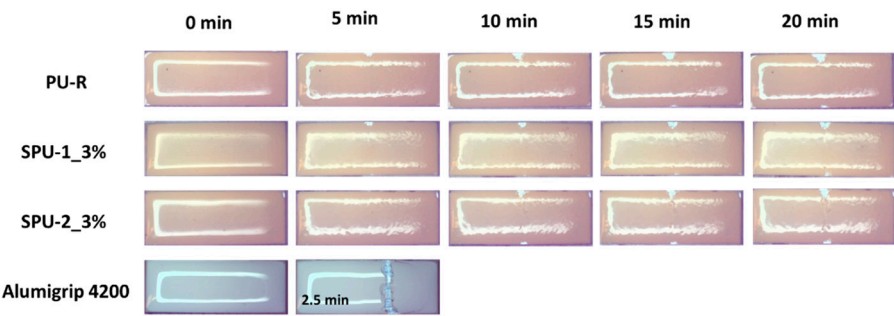

**Figure 7.** Water droplet erosion resistance of SPU coatings. Test conditions: 463 μm water droplets, impact velocity of 175 m/s, impact angle of 90°, impact frequency of 42,000 droplets per minute.

## 4. Conclusions

Novel surface-modifying polymers (SMPs) were designed, synthesized and formulated into the erosion resistant polyurethane coating PU-R to produce highly hydrophobic SPU coatings with superior erosion durability. The SPU coatings had high solid contents of ca. 58 wt.% (based on total resin solid) and could be easily applied by spraying or solution casting. Drying and curing of the coatings at ambient conditions produced elastomeric coatings with high water contact angles (100–115°), high mechanical strength (19–27 MPa), high elongation at break (640–730%) and low tensile set (20–35%). Reduced ice adhesion strength in the range of 220–400 kPa was measured for the SPU coatings using a static push rig test at −14 °C. Repeated icing/ice removal cycles led to no substantial increase in the ice adhesion strength. The ASTM G76 and G73 standard tests demonstrated excellent erosion resistance against both high-speed solid particles and water droplets for the SPU coatings. Thus, the combination of high erosion durability, high mechanical strength, high surface hydrophobicity and reduced ice adhesion strength makes the SPU coatings attractive for ice protection of leading edge surfaces of fast moving aerodynamic structures, such as aircraft wings and helicopter rotor blades.

**Author Contributions:** Conceptualization, N.S. and A.B.; methodology, N.S. and A.B.; validation, N.S.; investigation, N.S.; resources, A.B.; writing—original draft preparation, N.S.; writing—review and editing, A.B.; project administration, A.B. All authors have read and agreed to the published version of the manuscript.

**Funding:** This research was funded by the Integrated Aerial Mobility program and the Aerospace Future Initiative Program of the National Research Council Canada, and the Canadian Department of National Defence–Royal Canadian Navy.

**Institutional Review Board Statement:** Not applicable.

**Informed Consent Statement:** Not applicable.

**Data Availability Statement:** No extra supporting information is provided.

**Conflicts of Interest:** The authors declare no conflict of interest. The funders had no role in the design of the study; in the collection, analyses, or interpretation of data; in the writing of the manuscript.

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
