# Peer review of "Erosion Resistant Hydrophobic Coatings for Passive Ice Protection of Aircraft"

_applsci, doi:10.3390/app12199589_

Round 1
Reviewer 1 Report
The work relates to making anti-icing coatings that can be potentially used on aeronautic surfaces. The experiments appear to be well-designed on most part, results are adequately discussed, but some improvements are warranted that could strengthen the paper.
1. The IAS values of coatings is not particularly impressive. The smallest value is ~220kPa which is just a 3x reduction over the base material. For a coating to be designated as ‘icephobic’, some authors have suggested that IAS should be <10kPa (see references provided below). So, wouldn’t these coatings only provide limited benefit? Or in other words, would there be a real motivation to use these coatings when other materials have shown much better performance?
2. Figure8 is hard to understand. The images look almost transparent, and I am unsure what we are supposed to see in them?
3. For Fig7 and Fig8: just providing measurements on how much erosion takes place is insufficient. Can the authors provide other measurements wherein they may have measured IAS and contact angle (advancing/receding) on these surfaces AFTER the erosion tests?
4. Figures need improvements.
· Some figures can be combined to make it easier for readers to absorb the information. For example, combine Fig2-5 can all be combined in the same figure as they all relate to different types of characterization of the coatings.
· The font size of numbers in XY axis in all the figures should be the same and at-least 7pt to make them readable. The XY captions should be at-least 8pt.
· Remove the minor grey scale lines. These make figures look ugly.
5. The authors should include some additional up-to-date literature that has studied icing on coated surfaces such as:
· Golovin, K. et al. Designing durable icephobic surfaces. Sci. Adv. 2, e1501496, doi:10.1126/sciadv.1501496 (2016).
· Golovin, K., Dhyani, A., Thouless, M. D. & Tuteja, A. Low-interfacial toughness materials for effective large-scale deicing. Science 364, 371-375, doi:10.1126/science.aav1266 (2019).
· He, Z. et al. Bioinspired Multifunctional Anti-icing Hydrogel. Matter 2, 723-734, doi:10.1016/j.matt.2019.12.017 (2020).
· Ma, L., Zhang, Z., Gao, L., Liu, Y. & Hu, H. Bio‐Inspired Icephobic Coatings for Aircraft Icing Mitigation: A Critical Review. Progress in Adhesion and Adhesives 6, 171-201 (2021).
· Chatterjee, R., H. Bararnia, and S. Anand, A Family of Frost-Resistant and Icephobic Coatings. Advanced Materials, 2022. 34(20): p. 2109930.
Author Response
Response to Reviewer 1 Comments
Comment 1: The IAS values of coatings is not particularly impressive. The smallest value is ~220kPa which is just a 3x reduction over the base material. For a coating to be designated as ‘icephobic’, some authors have suggested that IAS should be <10kPa (see references provided below). So, wouldn’t these coatings only provide limited benefit? Or in other words, would there be a real motivation to use these coatings when other materials have shown much better performance?
Answer: We agree that the coating materials reported do not have the lowest ice adhesion strength required for passive ice protection. However, the significance of the study is more about adding icephobic functionality to a highly erosion resistant coating for leading edge protection, rather than pursuing high icephobicity alone. As far as we know, there is no erosion protective coatings exhibiting low ice adhesion strength. Neither there is an icephobic coating having both extremely low ice adhesion and good durability against high-speed impact of particulate matters. The combination of excellent erosion resistance and reduced ice adhesion strength makes the reported coatings unique. Certainly, there is a need to further improve the icephobicity for passive ice protection of aircraft.
Comment 2: Figure8 is hard to understand. The images look almost transparent, and I am unsure what we are supposed to see in them?
Answer: Figure 8 (now Figure 7) shows the erosion damage caused by high-speed impact of water droplets. Although the coatings are transparent, the occurrence and evolution of coating damages with time can still be clearly seen. In particular, when compared to the commercial reference coating, where total material removal was observed at the impact area at 2.5 min of testing, the coatings of this study showed significantly improved water droplet erosion resistance.
Comment 3: For Fig7 and Fig8: just providing measurements on how much erosion takes place is insufficient. Can the authors provide other measurements wherein they may have measured IAS and contact angle (advancing/receding) on these surfaces AFTER the erosion tests?
Answer: It would be ideal to be able to measure ice adhesion strength and water contact angles for the eroded samples. However, this cannot be done with the current set up. For sand erosion test, a beam of alumina particles was directed to the coating at an impingement angle of 30 deg, where the polymer coatings experience the most severe erosion damage. After the test, depending on the degree of erosion, a crater (ca. 10 mm, diameter) with different depth will be formed, which makes the measurement of IAS and contact angles impossible. For water droplet erosion test, the impacted area is a line of about 2 mm wide, which is too small for IAS and contact angle (advancing/receding) measurement. Having said that, water contact angles were measured for a similar but different hydrophobic coating that was subjected to sand blasting across the whole surface using a hand-held sand blaster. The static water contact angle increased from ca. 110 deg (before) to ca. 140 (after). However, due to the fact it is not possible to control the sand loading, impact speed and impact angle, the test was only a qualitative observation.
Comment 4: Figures need improvements.
- Some figures can be combined to make it easier for readers to absorb the information. For example, combine Fig2-5 can all be combined in the same figure as they all relate to different types of characterization of the coatings.
Answer: Tried to combine all of them into one Figure but since there are 5 figures, it is difficult to arrange them. Therefore, Figure 2 and Figure 3 were combined since both are chemical characterizations, Figure 4 (now Figure 3) and Figure 5 (now Figure 4) were kept separated because one is thermal characterization and one is mechanical testing.
- The font size of numbers in XY axis in all the figures should be the same and at-least 7pt to make them readable. The XY captions should be at-least 8pt.
Answer: All the font size for the XY numbers and titles were enlarged.
- Remove the minor grey scale lines. These make figures look ugly.
Answer: All grey scale lines were removed from the figures.
Comment 5: The authors should include some additional up-to-date literature that has studied icing on coated surfaces such as:
- Golovin, K. et al. Designing durable icephobic surfaces. Sci. Adv. 2, e1501496, doi:10.1126/sciadv.1501496 (2016).
- Golovin, K., Dhyani, A., Thouless, M. D. & Tuteja, A. Low-interfacial toughness materials for effective large-scale deicing. Science 364, 371-375, doi:10.1126/science.aav1266 (2019).
- He, Z. et al. Bioinspired Multifunctional Anti-icing Hydrogel. Matter 2, 723-734, doi:10.1016/j.matt.2019.12.017 (2020).
- Ma, L., Zhang, Z., Gao, L., Liu, Y. & Hu, H. Bio‐Inspired Icephobic Coatings for Aircraft Icing Mitigation: A Critical Review. Progress in Adhesion and Adhesives 6, 171-201 (2021).
- Chatterjee, R., H. Bararnia, and S. Anand, A Family of Frost-Resistant and Icephobic Coatings. Advanced Materials, 2022. 34(20): p. 2109930.
Answer: all missing references are added.
Reviewer 2 Report
Authors presents very interesting results and valuable analysis. Generally, conclusions are sound and justified by the data. Before accepting I recommend to proof the assignment the IR band at around 2200 cm-1 to vibrations of the NCO group.
Author Response
Response to Reviewer 2 Comments
Comment: Authors presents very interesting results and valuable analysis. Generally, conclusions are sound and justified by the data. Before accepting I recommend to proof the assignment the IR band at around 2200 cm-1 to vibrations of the NCO group.
Answer: Yes, it is confirmed that NCO has a characteristic IR absorption band at 2200-2300 cm-1, which has been used to monitor the degree of reaction. A reference can be: Xu, Li; Li, C.; Ng, K.Y.S. In-situ monitoring of urethane formation by FTIR and Raman spectroscopy. J. Phys. Chem. A 2000, 104, 3952-3957.
Reviewer 3 Report
The article deals with a hydrophobic material based on PU-R mixed with specific surface modifying polymers showing a degree of icephobicity. It describes preparation and synthesis of the above materials. Experimental part includes measurement of hydrophobicity via static and dynamic contact angle and icephobicity via an experimental rig. Erosion resistance and chemical composition are included as well.
The paper is well organized and clear with the exception of the chapter "Results and Discussion" containing the preparation of the specimens. This should probably be present in the chapter "Materials and Methods". Also, I would think about dividing the chapter of "Results nad Discussion" into sub chapters by the used methods.
Since the problematics of icephobicity is still a big topic, any new data on icephobic materials are important and useful.
However, I miss some information:
1) In introduction, principles and reasons of ice adhesion are missing. There should be a brief description of influence of modulus of elasticity on ice adhesion and of intermolecular forces between ice and solid which explain the suitability of elastomers regarding the icephobicity (low dielectric constant and surface energy).
2) I miss any background on why authors have used these materials. Regarding this, comparison with other elastomeric materials and their icephobic properties is missing. Values of ice adhesion reported in the article (220-400 kPa) are on the edge to be called "icephobic". The critical values of adhesion to ice in recent literature start from 100 kPa and less.
3) Ice adhesion measurement configuration as seen in Fig. 1 point out at the fact that the measurement process does not operatee in a pure shear mode. The contact point of the pushing rod and an ice block is not at the ice/sample surface interface. That is, of course, virtually impossible. However, if the distance from the contact point to the interface is too high a momentum or a torque is produced. Then a mixture of shear and tensile load is created. This torque lifts the side of contact of ice and rod and presses down the block on the opposite side. The block is then pressed into the material thus increasing the force neccesary to slide theice block. This could explain relatively high values of measured ice adhesion. This height distance must be presented. Otherwise, the results are not reproducible and comparable with any other research works. Moreover, the results of ice adhesion in this combined load mode can be significantly different from the results obtained in a pure shear mode. I recommend reading the article of "Shear-induced adhesive failure of a rigid slab in contact with a thin confined film by M.K. Chaudhury and K.H. Kim, 2007".
3) The data on loading rate or displacement velocity of the pushing rod during ice adhesion testing must be present. Here, again, I personally recommend reading the article above adressing the phenomenon of ice block just sliding but still adhering to elastic materials at low speeds but properly delaminating at greater speeds. Not necessary to incorporate it into the article.
4) The erosion is a very important mode of aircraft propellers wear. But when speaking about the mechanical wear resistance, any information on hardness should be present (Shore hardness tester or nanoindentation).
5) How many samples of a given material were used to measure ice adhesion?
6) Jus a quick note - regarding the conclusion, I would be very careful to call these materials highly attractive for fast moving aerodynamics structures. In case of the structure being stationary, the water rain drops will probably spread in time on the surface and freeze (since the water contact angles were experimentally measured immediatelly after 0,5 s). Then with such a low height of the ice formed, the force to remove them will be high since the drops will maintain "aerodynamic" flat shape. It will be up to an field test to confirm the suitability of the materials.
Author Response
Response to Reviewer 3 Comments
The article deals with a hydrophobic material based on PU-R mixed with specific surface modifying polymers showing a degree of icephobicity. It describes preparation and synthesis of the above materials. Experimental part includes measurement of hydrophobicity via static and dynamic contact angle and icephobicity via an experimental rig. Erosion resistance and chemical composition are included as well.
The paper is well organized and clear with the exception of the chapter "Results and Discussion" containing the preparation of the specimens. This should probably be present in the chapter "Materials and Methods". Also, I would think about dividing the chapter of "Results nad Discussion" into sub chapters by the used methods.
Answer: All the unnecessary description of sample preparations and test conditions were removed from the “Results and Discussion” section and added to the “Materials and Method”
Since the problematics of icephobicity is still a big topic, any new data on icephobic materials are important and useful.
However, I miss some information:
Comment 1: In introduction, principles and reasons of ice adhesion are missing. There should be a brief description of influence of modulus of elasticity on ice adhesion and of intermolecular forces between ice and solid which explain the suitability of elastomers regarding the icephobicity (low dielectric constant and surface energy).
Answer: Icephobic coatings based on different principles were briefly described in the introduction and their key shortcomings were pointed out. It should be noted that although the coatings reported here are elastomeric, they are different from those soft, low-shear-modulus icephobic coatings. The coatings of this study have a shore A hardness of about 70, which won’t produce significant interface defects under shear to cause ice detachment, like those observed for soft rubbers reported by Golovin et al. (Golovin, K., Dhyani, A., Thouless, M. D. & Tuteja, A. Low-interfacial toughness materials for effective large-scale deicing. Science 364, 371-375, doi:10.1126/science.aav1266, 2019).
Comment 2: I miss any background on why authors have used these materials. Regarding this, comparison with other elastomeric materials and their icephobic properties is missing. Values of ice adhesion reported in the article (220-400 kPa) are on the edge to be called "icephobic". The critical values of adhesion to ice in recent literature start from 100 kPa and less.
Answer: The significance of the study is to achieve coatings that have both excellent erosion resistance and reduced ice adhesion strength. Leading edge surfaces of aircraft are susceptible to erosion damage caused by high-speed impact of solid particles and rain droplets. There is no commercial coating is available on the market to provide good enough erosion protection. The matrix coating (PU-R) reported here showed outstanding erosion resistance against both sand particles and rain droplets. On the other hand, the forward-facing leading edge surfaces are also more prone to icing than other area. It is therefore make sense to add icephobic functionality to the erosion resistance coating. Given the fact that erosion durability has higher priority than icephobicity, at least for aircraft applications, the added icephobicity should not be realized at the cost of durability. This somehow limits the means to lower the ice adhesion strength. It is agreed that the reported ice adhesion strength is not low enough for passive ice protection. But it has been lowered to the same level with those of silicones and PTFE without sacrificing erosion resistance. Certainly, further work need to be done to improve the icephobicity for the purpose of passive ice protection of aircraft.
Comment 3: Ice adhesion measurement configuration as seen in Fig. 1 point out at the fact that the measurement process does not operatee in a pure shear mode. The contact point of the pushing rod and an ice block is not at the ice/sample surface interface. That is, of course, virtually impossible. However, if the distance from the contact point to the interface is too high a momentum or a torque is produced. Then a mixture of shear and tensile load is created. This torque lifts the side of contact of ice and rod and presses down the block on the opposite side. The block is then pressed into the material thus increasing the force neccesary to slide theice block. This could explain relatively high values of measured ice adhesion. This height distance must be presented. Otherwise, the results are not reproducible and comparable with any other research works. Moreover, the results of ice adhesion in this combined load mode can be significantly different from the results obtained in a pure shear mode. I recommend reading the article of "Shear-induced adhesive failure of a rigid slab in contact with a thin confined film by M.K. Chaudhury and K.H. Kim, 2007".
Answer: We fully agree with the comment and did consider this when designing the test rig. It is true that the “pusher shaft” is not positioned perfectly at the ice-coating interface, but the height of the pusher to the coating surface is only about 5 mm. Compared to the size of the ice block (25.4 mm wide x 25.4 mm deep x 7.8 mm high) or the size of the whole aluminum mold (75 mm x 75 mm x 7.8 mm), this height is not expected to exert significant torque or vertical pressure on the far end. In addition, our coatings are ‘relatively’ hard with a shore A hardness of about 70, much higher than silicones that typically have shore A hardness lower than 40. They are less sensitive to vertical pressure, if there is any, caused by pushing at a height. Of course, a pure shear test would be ideal and we will continue to improve the test rig on that end.
Comment 4: The data on loading rate or displacement velocity of the pushing rod during ice adhesion testing must be present. Here, again, I personally recommend reading the article above adressing the phenomenon of ice block just sliding but still adhering to elastic materials at low speeds but properly delaminating at greater speeds. Not necessary to incorporate it into the article.
Answer: The shear load was applied to the pusher by manually turning the threaded handle in the back. The loading rate will depends on the resistance felt by the hand and it should be in the range of 50-100 N/second. The displacement rate is hard to estimate. The coatings reported here are unlike silicones where ice slides on the surface. Ice block on our coatings detached suddenly and completely under a critical shear force.
Comment 5: The erosion is a very important mode of aircraft propellers wear. But when speaking about the mechanical wear resistance, any information on hardness should be present (Shore hardness tester or nanoindentation).
Answer: Shore A hardness was measured and added to the context.
Comment 6: How many samples of a given material were used to measure ice adhesion?
Answer: For most of the coatings, only one sample was measured for the ice adhesion strength, but the measurement was repeated multiple times.
Comment 7: Jus a quick note - regarding the conclusion, I would be very careful to call these materials highly attractive for fast moving aerodynamics structures. In case of the structure being stationary, the water rain drops will probably spread in time on the surface and freeze (since the water contact angles were experimentally measured immediatelly after 0,5 s). Then with such a low height of the ice formed, the force to remove them will be high since the drops will maintain "aerodynamic" flat shape. It will be up to an field test to confirm the suitability of the materials.
Answer: Water may ‘spread’ on the coating surface due to contamination not on clean surfaces. Some visual observations were made for coatings rubbed with dirt and oil. The static contact angle reduced from >90 deg to <90 deg and typically in the range of 60 – 90 deg after contamination. Water will freeze no matter it is in moving or stationary. In our case, the height of the ice is not an important factor since interface defect under shear is not a consideration. The reason we state our coatings are “high attractive for fast moving structures” is because such structures need both erosion durability and low ice adhesion and our coatings showed both excellent erosion resistance and reduced ice adhesion. Nevertheless, the word “highly” is removed since icephobicity of the coatings is still to be improved.
Round 2
Reviewer 1 Report
The authors have addressed my concerns. Happy to recommend its acceptance.
Reviewer 3 Report
In future, try to incorporate the answers for reviewer´s questions into the article at least in some very short way.
Otherwise, good job.
Good luck in your other papers